# A Pilot Study in Sweden on Efficacy of Benzylpenicillin, Oxytetracycline, and Florfenicol in Treatment of Acute Undifferentiated Respiratory Disease in Calves

**DOI:** 10.3390/antibiotics9110736

**Published:** 2020-10-26

**Authors:** Virpi Welling, Nils Lundeheim, Björn Bengtsson

**Affiliations:** 1Farm and Animal Health, Kungsängens Gård, SE-753 23 Uppsala, Sweden; virpi.welling@gardochdjurhalsan.se; 2Department of Animal Breeding and Genetics, Swedish University of Agricultural Sciences, Box 7023, SE-750 07 Uppsala, Sweden; nils.lundeheim@slu.se; 3Department of Animal Health and Antimicrobial Strategies, National Veterinary Institute, SE-751 89 Uppsala, Sweden

**Keywords:** cattle, respiratory disease, treatment, benzylpenicillin, oxytetracycline, florfenicol

## Abstract

Bovine respiratory disease (BRD) is a major indication for antibiotic treatment of cattle worldwide and some of the antibiotics used belong to classes of highest priority among those listed by WHO as critically important for human medicine. To preserve the efficacy of “newer” antibiotics, it has been suggested that “older” drugs should be revisited and used when possible. In this pilot study, we evaluated the efficacy of benzylpenicillin (PEN), oxytetracycline (OTC), and florfenicol (FLO) for treatment of naturally occurring BRD on two farms raising calves for slaughter. Farm personnel selected calves for enrolment, assigned calves to one of the three regimens in a systematically random manner, treated the calves, and registered the results. Overall, 117 calves were enrolled in the study. Nineteen calves relapsed in BRD before slaughter and were retreated (16.2%) and three died (2.6%). For PEN, treatment response rates after 30 days, 60 days, and until slaughter were 90.2%, 87.8%, and 80.5%, respectively; for OTC, 90.0%, 85.0%, and 85.0%, respectively; and for FLO, 86.1%, 83.3%, and 77.8%, respectively. There were no statistically significant differences in relapse, mortality, or response rates between the three treatment regimens. This indicates that PEN, OTC, and FLO were equally effective for treatment of BRD but the results need to be confirmed in a more elaborate study with a higher statistical power. The findings support the current recommendations from the Swedish Veterinary Association and the Medical Products Agency to use benzylpenicillin as a first line antibiotic for treatment of calves with undifferentiated respiratory disease in Sweden. Due to differences in the panorama of infectious agents and presence of acquired antibiotic resistance, the findings might not be applicable in other geographical areas.

## 1. Introduction

Bovine respiratory disease (BRD) is recognized worldwide as a common and persistent problem in cattle raised intensively for meat production [1,2,3,4]. BRD has a multifactorial background that includes infectious agents and environmental factors but also the immunological and general status of the animals [5,6]. The pathogenesis and clinical presentation of BRD varies depending on which infectious agents and predisposing factors are present in a herd, but in general, viral infections of the respiratory tract precede secondary bacterial infections [7]. The bacteria commonly involved often reside in the upper airways of healthy calves and include *Pasteurella multocida*, *Mannheimia haemolytica*, *Histophilus somni*, and *Mycoplasma bovis* [7,8]. Due to the complex background of BRD, morbidity and mortality vary between herds, but both can be very high [9,10] leading to animal welfare problems and economic losses for the farmers [6,11].

There is no curative therapy for viral respiratory infections available for use in cattle and treatment of manifest BRD therefore relies on antibiotics to control secondary bacterial infections of the lower respiratory tract and NSAIDs to alleviate the inflammatory response. Worldwide, antibiotics are also used extensively for prophylactic or metaphylactic medication to prevent or curb outbreaks of BRD in groups of calves or growing cattle [9,12]. Thus, BRD is the major indication for antibiotic treatment of cattle [9,13] and the risk of emerging antibiotic resistance from this has been emphasized [14]. Some of the antibiotics used to control BRD, for example, fluoroquinolones, third generation cephalosporins, and macrolides, belong to antibiotic classes of highest priority among those listed by WHO as critically important for human medicine (CIA) [15].

Due to concerns for human and animal health from emerging resistance, WHO recently suggested that CIAs should not be used for treatment of food-producing animals if alternatives are available [16]. It has also been suggested that to preserve the efficacy of “newer” antibiotics “older” drugs should be re-investigated and used when possible [17]. In Sweden, benzylpenicillin is recommended by the Swedish Veterinary Association [18] and by the Medical products agency [19] as first line treatment of cattle with lower respiratory tract infections. The rationale for using this “old” antibiotic is a lower risk for emergence of resistance from the use of the narrow-spectrum benzylpenicillin than the broad-spectrum antibiotics available for treatment of BRD in Sweden, i.e., oxytetracycline, florfenicol, enrofloxacin, amoxicillin, tulathromycin, gamithromycin, trimethoprim-sulphonamide, and ceftiofur.

Penicillin resistance in respiratory pathogens from calves is uncommon in Sweden [20] and *M. bovis* has hitherto been found only rarely [21]. In most herds, the efficacy of benzylpenicillin should therefore not be compromised by these factors. The efficacy of benzylpenicillin for the treatment of BRD has, however, never been evaluated in Swedish cattle herds and studies are also scarce in the scientific literature [10,22]. Information on performance of benzylpenicillin in Swedish herds would be valuable to uphold, or reevaluate, the current recommendation. The aim of this pilot study was therefore to evaluate the efficacy of benzylpenicillin (PEN) for treatment of calves with naturally occurring BRD in comparison with two other antibiotics used in Sweden, i.e., oxytetracycline (OTC) and florfenicol (FLO).

## 2. Results

### 2.1. Descriptive Data

The study was designed to include three farms and 180 calves in total, but one farm dropped out at an early stage. On the remaining two farms (H and S), 120 calves were enrolled in the study. Due to uncertainties in farm records, three calves were excluded, one calf on farm S and two calves on farm H, leaving 117 calves for final evaluation (Table 1). Of these, 59 calves were from farm S treated between February and April 2016 and 58 from farm H treated between February 2016 and May 2017. The stipulated order of assignment to treatment regime (PEN, OTC, FLO) at enrolment was upheld throughout the study but with minor deviations on both farms.

The mean age of the calves at enrollment was 48.2 days and did not differ significantly between treatment regimens, but calves on farm H were significantly older (53.4 days) at enrollment than calves on farm S (42.4 days) (*p* = 0.004) (Table 1). The mean rectal temperature at enrolment was 39.8 °C, the mean age at slaughter 539 days, and the mean carcass weight at slaughter 318.4 kg (Table 1). None of these parameters differed significantly (*p* > 0.05) between treatment regimens or between the two farms (Table 1).

Of the 117 calves, 19 (16.2%) relapsed in respiratory disease after first treatment and were retreated (Table 1, Appendix A). The relapse rate, from first treatment up to slaughter, was 14.6% for PEN, 12.5% for OTC, and 22.2% for FLO. The relapse rate did not differ between regimens (*p* > 0.05) but was significantly higher on farm S (28.8%) than on farm H (3.5%) (*p* = 0.0002) (Table 1). Three calves relapsed twice, one of these was first treated with OTC and two with FLO (Table 1). Three calves died (2.6%), two on farm S and one on farm H (Table 1, Appendix A). All three calves were first treated with PEN and two died during the first treatment and one during retreatment with PEN 88 days after the first treatment. There was no statistically significant difference in mortality between treatment regimens or between farms (*p* > 0.05).

### 2.2. Efficacy Parameters

Most calves, 115 (98.3%), fulfilled the criteria for a positive temperature reaction (TEMP) (Table 2, Appendix A). Response rates for the three treatment regimens at 30 days (RESP_30_), 60 days (RESP_60_), and until slaughter (RESP_tot_) were 90.2%, 87.8%, and 80.5%, respectively, for PEN; 90.0%, 85.0%, and 85.0%, respectively, for OTC; and 86.1%, 83.3%, and 77.8%, respectively, for FLO (Table 2). There were no statistically significant differences in response rates between treatment regimens (*p* > 0.05) (Table 2). There was also no difference in mean RESP_30_ between farms (*p* > 0.05), but mean RESP_60_ was significantly lower on farm S (76.3%) than on farm H (94.8%) (*p* = 0.0044) and the mean RESP_to_t was also significantly lower on farm S (67.8%) than on farm H (94.8%) (*p* = 0.0002) (Table 2). The perceived treatment effect at five days (PTE) was scored as “Good” for 91.2% of the calves and as “Poor” for 8.8% and did not differ between treatment regimens or between farms (Table 2). The mean average daily live weight gain from birth to slaughter (ADG) was 1006 grams/day and did not differ significantly between treatment regimens or between farms (*p* > 0.05) Table 2).

## 3. Discussion

Benzylpenicillin is one of the oldest antibiotics used for treatment of BRD and has to a great extent been replaced by newer substances with a broader antibacterial spectrum [23]. However, in this study, we observed no difference in efficacy of benzylpenicillin, oxytetracycline, or florfenicol for treatment of naturally occurring BRD in Swedish calves raised for meat production. The response rates (RESP) at 30 days, 60 days, and until slaughter were 90.2%, 87.8%, and 80.5%, respectively, for PEN; 90.0%, 85.0%, and 85.0%, respectively, for OTC; and 86.1%, 83.3%, and 77.8%, respectively, for FLO and did not differ between the three regimens (*p* > 0.05). Moreover, the perceived treatment effects (PTE) scored by farmers at five days were high. In 91.2% of the calves, the effect was scored as good and did not differ between treatment regimens. A response rate of 80–85% after the first treatment is considered acceptable for BRD in feedlot cattle [23], and all three regimens evaluated in this study can therefore be considered adequate.

The overall response rates at 30 days and 60 days after first treatment in the present study are similar to response rates of about 85% at 28 days observed after treatment of weaned feedlot cattle with benzylpenicillin, oxytetracycline, or trimethoprim/sulphonamide reported by Bateman et al. [10] and higher than rates of about 50–60% at 60 days after treatment with these antibiotics reported by Mechor et al. [22]. Furthermore, the overall response rate up until slaughter in the present study (81.2%) was higher than reported for treatment of feedlot cattle with enrofloxacin, about 60–70% [24], or tilmicosin (34–67%) [25,26,27], and within the interval of success rates reported for florfenicol (23–97%) [25,26,27], tulathromycin (53–88%) [24,26,27], trimethoprim/sulphonamide (77%) [28], and ceftiofur (90%) [28]. Moreover, in a review of randomized control trials on BRD treatment with various antibiotics, the median success rate was 71% in treated animals and 24% in untreated controls [29]. Notably, in mixed treatment comparison meta-analyses of results from BRD-treatment studies performed in North American feedlots, tulathromycin ranked the highest and oxytetracycline the lowest with florfenicol ranking as number four on the list of 12 antibiotics evaluated; penicillin was not included in the analysis [30].

To compare success rates between BRD treatments studies should, however, be made cautiously due to possible differences in study design, e.g., study populations, dosing regimens, case definition, and the criteria used for evaluation of success or failure [29]. Additionally, the outcome of antimicrobial therapy for BRD is most likely influenced by the disease challenge in a herd [31], and extrapolation of results between different settings should be made cautiously. In Sweden, some infectious agents of importance in BRD elsewhere are not present, for example, BVDV and BHV-1 [32] and *M. haemolytica*, *H. somni*, and *M. bovis* are less often diagnosed [19,21]. The bacterial pathogen mainly isolated from calves with BRD in Sweden is *P. multocida*, and antimicrobial resistance to benzylpenicillin, oxytetracycline, or florfenicol in this bacterial species is uncommon [19,20]. The good performance of the antibiotics studied was therefore probably to some extent due to a relatively low disease challenge. The results of this study might therefore not be relevant in other settings where, for example, *M. bovis* is more common, such as North American feedlots [2], or where the occurrence of acquired antibiotic resistance is higher.

The total mortality in the study (2.6%) was lower than the overall mortality, including fatalities unrelated to BRD, on farm S (3.3%) and farm H (4.5%) in 2016. Notably, the three calves that died in the study were all treated with benzylpenicillin but there were no statistically significant differences in mortality between treatment regimens. Since no post-mortem examinations were performed, it is not known if these calves had comorbidities unrelated to BRD, for example, the calf that died about three months after the first treatment. Still, this finding is intriguing and warrants clarification in a more elaborate study.

The frequency of BRD treatment was similar on the two farms, at about 20%, but the overall response rates RESP_60_ and RESP_tot_ were significantly lower on farm S (76.3% and 67.8%, respectively) than on farm H (94.8% and 94.8%, respectively). The reasons for the difference are unclear, but farm S yearly purchases and raises about ten times more animals than farm H, and a greater flow of animals likely influences the spectrum of infectious agents present. The immunological and general status of the animals is also possibly more diverse on a larger farm and the supervision and management of diseased animals more complicated. This has probably an impact on the outcome of BRD treatments.

All calves in the present study were treated with NSAID (meloxicam) in conjunction with antibiotic therapy because this is recommended in the treatment of BRD in Sweden [33]. The initial drop in rectal temperature in all but 2 of the 117 calves was probably to some extent due to the antipyretic effect of meloxicam in agreement with the conclusions of a review on ancillary treatment of BRD by Francoz et al. [34]. These authors also concluded that NSAIDs may be beneficial from an animal welfare perspective and possibly also reduce lung lesions at slaughter whereas beneficial effects on clinical signs at the end of treatment and on productivity are not documented. To what extent the ancillary treatment influenced the overall response in the present study cannot be evaluated, but since all calves were treated with meloxicam, any impact on the conclusions regarding relative efficacy of the three regimens is likely to be small.

Respiratory infections in calves are known to negatively impact the productivity [5,35] but this aspect was not directly evaluated in the current study. However, the mean age at slaughter (539 days), the mean carcass weight (318.4 kg), and the average daily live weight gain (1066 grams/day) of the treated calves were higher than the national average for dairy breed bull calves in Sweden, 615 days, 334 kg, and 980 grams/day, respectively [36], and did not differ between the three treatment regimens or the two farms. This indicates a successful management on the two farms but also that all the three treatment regimens restored the long-term productivity of the calves.

A main limitation of this study is the small number of animals and farms included. The study was originally designed to include three farms, but the drop-out of one farm led to a loss of statistical power in the evaluation of the study. With the current number of observations, using a significance level of 0.05 and a statistical power of 80%, only a difference of about 20% between treatment regimens could be detected. Other limitations are that calves were selected for treatment by farm personnel based on visual inspection and rectal temperature. Selection in this manner inevitably leads to underdiagnosis of calves needing antibiotic therapy [6,9] but also to a selection of calves that would have recovered spontaneously [29]. Inclusion of calves that would have recovered spontaneously would overestimate the efficacy of the treatment regimens studied. However, the proportion of calves that would have recovered spontaneously is probably similar for the three regimens and the comparison of efficacy between regimes would remain unaffected. More concerning is that farm personnel, also on visual inspection and temperature recordings, decided which calves should be retreated and thereby considered relapses which directly impacts response rates. Furthermore, farm personnel were not blinded with respect to treatment regimen in individual calves and their perception of the efficacy of the antibiotics used could have biased the likelihood of selecting calves for retreatment. Another limitation is that calves that died not were submitted for post-mortem investigation and the causes of the fatalities are not known and could be unrelated to BRD. Moreover, isolation of respiratory tract pathogens or susceptibility testing of relevant isolates were not performed.

Despite these limitations, we consider it valuable to share the results obtained in this pilot study, although the conclusions should be confirmed in studies accounting for these constraints and with a higher statistical power.

## 4. Materials and Methods

### 4.1. Study Design

The study was designed to evaluate the efficacy of benzylpenicillin, oxytetracycline, and florfenicol in treatment of naturally occurring BRD in Swedish herds raising calves for slaughter (for an overview see Appendix A). The intention was to perform a non-inferiority pilot study using three herds selected by convenience and with a minimum of intervention in farm routines and extra work for farm personnel. Initially, three farms were enrolled in the study and a total of 60 calves were to be treated with each of the three regimens. At an expected treatment success rate of 85%, this would have given the study a power of 80% to detect a true difference in success rates between regimens of about 15% at a significance level of 5% (Sealed Envelope LTD. 2012). This was considered sufficient for a preliminary evaluation of the performance of the studied antibiotics, but, unfortunately, one of the herds dropped out of the study which reduced the non-inferiority limit to 20%.

### 4.2. Farms and Animals

The study was conducted in 2016 and 2017 on two farms (S and H) that purchased unweaned calves from dairy farms and raised them for slaughter at an age of about 18 months. Calves were purchased and received to both farms in batches over the whole year and kept in groups of 10–15 animals. Calves were fed milk substitutes, and gradually concentrates and silage, until weaned at a bodyweight of 90–100 kg and an age of about 8–12 weeks. After weaning, calves were mixed on both farms in larger groups of 20–30 animals and raised to slaughter on a mixed ration of concentrates and silage. On both farms, calves commonly contracted respiratory disease 1–2 weeks after arrival. During 2016, farm S received 2528 calves emanating from 59 different farms. Of these calves, 530 (20.9%) were treated for BRD, 43 (1.7%) died or were euthanized before reaching a bodyweight of 100 kg, and 40 calves (1.6%) died or were euthanized in the period thereafter and up until slaughter. The same year, farm H received 268 calves emanating from one single dairy farm, and of these, 64 (23.9%) were treated for BRD, 4 (1.5%) died or were euthanized before reaching a bodyweight of 100 kg, and 8 (3.0%) calves died or were euthanized in the period thereafter. Prophylactic antibiotic treatment, or use of antibiotics for growth promotion, is not allowed in Sweden and accordingly not practiced on the farms.

### 4.3. Inclusion Criteria

On both farms, calves were visually inspected daily by farm personnel for signs of disease and rectal temperature was recorded for calves showing clinical symptoms. For this study, farm personnel were instructed to identify calves with signs of respiratory disease, including (I) forced breathing and/or cough, (II) purulent nasal and/or ocular discharge, (III) depressed attitude, and (IV) a rectal temperature of >39.5 °C. Calves fulfilling at least three of the criteria I–IV were enrolled in the study. These criteria for starting antibiotic treatment of calves with suspected respiratory infection were the same as those used on both farms also before the start of the study. Calves with comorbidities or calves older than 6 months were not eligible for enrollment.

### 4.4. Treatment Regimens

Calves enrolled in the study were treated according to one of the following three regimens according to the manufacturers recommendations—PEN: procaine benzylpenicillin 40 mg/kg BW IM, 5 doses 24 H apart (Penovet^®^ vet, Boehringer Ingelheim Animal Health, Copenhagen, Denmark); OTC: oxytetracycline 20 mg/kg BW IM, 2 doses 48 H apart (Engemycin^®^ vet, Intervet AB, Stockholm, Sweden); FLO: florfenicol 20 mg/kg BW, 2 doses 48 H apart (Florselect^®^ vet, Nordvacc Läkemedel AB, Hägersten, Sweden). Calves were assigned to a regimen in order of enrollment, where the first calf identified for treatment on a farm received PEN, the second OTC, and the third FLO. This order of treatments was to be repeated until a total of 60 calves had been treated on each farm. If farm personnel considered that the clinical response of a calf was unsatisfactory, they could change the treatment to one of the other two regimens. Calves relapsing in respiratory disease after completed initial treatment were again treated with one of the three regimens at the discretion of the farm personnel.

At the start of antibiotic therapy, all calves were treated with an NSAID given as a single subcutaneous dose of meloxicam at 0.5 mg/kg BW, (Metacam^®^, Boehringer Ingelheim Vetmedica, Malmö, Sweden).

### 4.5. Data Registered on Farm 

On enrollment (0 h), farm personnel registered calf identity, date, assigned treatment regimen, and rectal temperature. At 48 h, rectal temperature was again registered and at 120 h the perceived treatment effect (PTE) was scored by farm personnel (see below). Changes of an assigned treatment, relapse in respiratory disease after completed treatment, and case fatality up until slaughter were also registered. Due to practical constraints, the study could not be performed in a blinded manner at the farm level. Data on birth date of calves, live weight on arrival to the farm, and age at slaughter were available from farm records.

### 4.6. Data Registered at Slaughter

At slaughter, carcass weights of the calves were recorded by slaughterhouse personnel unaware of the treatment of individual calves.

### 4.7. Efficacy Parameters

Treatment efficacy was evaluated by the following parameters:TEMP (temperature): A positive reaction was a rectal temperature ≤39.5 °C and/or a drop by ≥1 °C 48 h after first treatment.RESP (response to treatment): A positive RESP was a positive reaction for the TEMP parameter (see above), no change of initial treatment and no relapse or fatality within 30 days (RESP_30_), 60 days (RESP_60_), or until slaughter (RESP_tot_).PTE (perceived treatment effect): Scored by farm personnel five days after first treatment as “Good” for a calf with noticeable improvements regarding clinical signs and general attitude, or “Poor” for a calf without noticeable improvements.ADG (average daily live weight gain from birth to slaughter).

### 4.8. Statistical Analyses

Possible differences in age and rectal temperatures at enrollment between calves enrolled in the three treatment regimen groups were evaluated by analysis of variance (PROC GLM) according to a statistical model including the fixed effects of treatment regimen (n = 3) and farm (n = 2).

To evaluate differences in efficacy between the three treatment regimens, the binary efficacy parameters TEMP, PTE, RESP_30_, RESP_60_, and RESP_tot_ were analyzed by logistic regression (PROC GLIMMIX) with a statistical model including the fixed effects of treatment regimen (n = 3) and farm (n = 2). Differences in total relapse and case fatality rates were analyzed with the same model. Differences between treatment regimens in age at slaughter, carcass weight, and average daily gain from birth to slaughter (ADG) were evaluated by analysis of variance (PROC GLM), with a statistical model that included the fixed effects of treatment (n = 3) and farm (n = 2).

Descriptive statistics was obtained using EXCEL, and statistical analyses were performed using the SAS software (SAS Inst. Inc., Cary, NC). A 5% level of significance was used to assess statistical differences. 

### 4.9. Ethics Approval

This study was approved by the regional ethical committee in Uppsala (DNR C 147/15).

## 5. Conclusions

The findings of this study indicate that PEN, OTC, and FLO were equally effective for treatment of undifferentiated BRD in calves, but the results need to be confirmed in a more elaborate study with a higher statistical power.

## Figures and Tables

**Table 1 antibiotics-09-00736-t001:** Descriptive data for 117 calves treated with procaine benzylpenicillin (PEN), oxytetracycline (OTC), or florfenicol (FLO) for respiratory disease on farm S and H.

		PEN			OTC			FLO			PEN & OTC & FLO		
Farm		S	H	S & H	S	H	S & H	S	H	S & H	S	H	S & H
No. of calves		20	21	**41**	21	19	**40**	18	18	**36**	59	58	**117**
Age when treated (days) ^a^	Mean (range)	42.6 (16–81)	51.2 (29–121)	**47.2** (16–121)	43.5 (10–88)	55.6 (25–107)	**49.6** (10–107)	40.7 (9–77)	53.6 (29–121)	**47.9** (9–121)	42.4 ** (9–88)	53.4 ** (25–121)	**48.2** (9–121)
Rectal temp. 0 h (°C)	Mean (range)	39.8 (38.4–41.5)	39.8 (38.4–41.3)	**39.8** (38.4–41.3)	40.1 (38.8–41.4)	39.6 (38.1–41.1)	**39.9** (38.1–41.4)	39.9 (37.1–41.3)	39.6 (37.5–41.0)	**39.8** (37.1–41.3)	39.9 (37.1–41.5)	39.7 (37.5–41.3)	**39.8** (37.1–41.5)
Rectal temp. 48 h (°C)	Mean (range)	38.4 (37.4–40.1)	38.4 (37.7–39.1)	**38.4** (37.4–40.1)	38.3 (36.3–40.4)	38.6 (37.5–39.9)	**38.4** (36.3–40.4)	38.2 (37.0–39.1)	38.3 (37.2–40.0)	**38.3** (37.0–40.0)	38.3 (36.3–40.4)	38.5 (37.2–40.0)	**38.4** (36.3–40.4)
Retreatment:													
<30 days	No.	0	2	**2**	2 ^c^	0	**2**	5 ^d^	0	**5**	7	2	**9**
30–60 days	No.	1	0	**1**	2	0	**2**	1 ^e^	0	**1**	4	0	**4**
>60 days	No.	3	0	**3**	1	0	**1**	2	0	**2**	6	0	**6**
Total	No.	4	2	**6**	4	0	**5**	6	0	**8**	17 **	2 **	**19**
Case fatality:													
<30 days	No.	1	1	**2**	0	0	**0**	0	0	**0**	1	1	**2**
30–60 days	No.	0	0	**0**	0	0	**0**	0	0	**0**	0	0	**0**
>60 days	No.	1 (day 90)	0	**1**	0	0	**0**	0	0	**0**	1	0	**1**
Total Case fatality	No	2	1	**3**	0	0	**0**	0	0	**0**	2	1	**3**
Age at slaughter (days) ^b^	Mean (range)	540 (450–597)	533 (482–574)	**536** (450–597)	544 (453–612)	526 (476–569	**535** (453–612)	552 (474–648)	539 (479–588)	**545** (474–648)	545 (450–648)	532 (476–588)	**539** (450–648)
Carcass weight (kg)	Mean (range)	311.8 (230–369)	326.8 (286–358)	**319.3** (230–369)	320.7 (260–364)	317.7 (287–356)	**319.2** (260–364)	315.0 (276–341)	318.1 (257–384)	**316.5** (257–384)	316.0 (230–369)	321.0 (257–384)	**318.4** (230–384)

^a^ Data for 8 calves missing (farm S: 2 PEN, 2 OTC, 4 FLO); ^b^ data for 22 calves missing (farm S: 4 PEN, 4 OTC, 5 FLO; Farm H: 4 PEN, 1 OTC, 4 FLO); ^c^ 1 calf retreated 66 days after first retreatment; ^d^ 1 calf retreated 99 days after first retreatment; ^e^ 1 calf retreated 59 days after first retreatment. Statistically significant differences between mean values indicated by ** (*p* < 0.01).

**Table 2 antibiotics-09-00736-t002:** Efficacy parameters for the treatment regimens procaine benzylpenicillin (PEN), oxytetracycline (OTC), and florfenicol (FLO) in 117 calves treated for respiratory disease on two farms, S and H. Percentage of calves fulfilling the criteria for the parameters: TEMP, RESP_30_, RESP_60_ and RESP_tot_; percentage of calves scored in the categories “Poor” or “Good” of parameter PTE; mean average daily live weight gain (ADG).

Farm			PEN			OTC			FLO			PEN & OTC & FLO		
			S	H	S & H	S	H	S & H	S	H	S & H	S	H	S & H
No. of calves			20	21	**41**	21	19	**40**	18	18	**36**	59	58	**117**
TEMP		% (no./total)	100 (20/20)	100 (21/21)	**100** (41/41)	90.5 (19/21)	100 (19/19)	**95.0** (38/40)	100 (18/18)	100 (18/18)	**100** (36/36)	96.6 (57/59)	100 (58/58)	**98.3** (115/117)
RESP_30_ (n = 117)		% (no./total)	95.0 (19/20)	85.7 (18/21)	**90.2** (37/41)	81.0 (17/21)	100 (19/19)	**90.0** (36/40)	72.2 (13/18)	100 (18/18)	**86.1** (31/36)	83.1 (49/59)	94.8 (55/58)	**88.9** (104/117)
RESP_60_ (n = 117)		% (no./total)	90.0 (18/20)	85.7 (18/21)	**87.8** (36/41)	71.4 (15/21)	100 (19/19)	**85.0** (34/40)	66.7 (12/18)	100 (18/18)	**83.3** (30/36)	76.3 ** (45/59)	94.8 ** (55/58)	**85.5** (100/117)
RESP_tot_ (n = 117)		% (no./total)	75.0 (15/20)	85.7 (18/21)	**80.5** (33/41)	71.4 (15/21)	100 (19/19)	**85.0** (34/40)	55.6 (10/18)	100 (18/18)	**77.8** (28/36)	67.8 *** (40/59)	94.8 *** (55/58)	**81.2** (95/117)
PTE (n = 113) ^a^	Poor	% (no./total)	11.1 (2/18)	10.0 (2/20)	**10.5** (4/38)	5.0 (1/20)	5.3 (1/19)	**5.3** (2/39)	5.5 (1/18)	16.7 (3/18)	**11.1** (4/36)	7.1 (4/56)	10.5 (6/57)	**8.8** (10/113)
	Good	% (no./total)	88.8 (16/18)	90.0 (18/20)	**89.5** (34/38)	95.0 (19/20)	94.7 (18/19)	**94.9** (37/39)	94.4 (17/18)	83.3 (15/18)	**88.9** (32/36)	92.9 (52/56)	89.5 (51/57)	**91.2** (103/113)
ADG (n = 99) ^b^		grams/day (range)	1039 (835–1266)	1104 (925–1204)	**1072** (835–1266)	1063 (887–1243)	1185 (936–1239)	**1074** (887–1243)	1033 (784–1210)	1063 (804–1189)	**1048** (784–1210)	1046 (784–1266)	1086 (804–1239)	**1066** (784–1266)

^a^ Data for 4 calves missing (farm S: 2 PEN, 1 OTC; farm H: 1 PEN); ^b^ data for 18 calves missing (farm S: 3 PEN, 2 OTC, 4 FLO; farm H: 4 PEN, 1 OTC, 4 FLO). Statistically significant differences between mean values indicated by ** (*p* < 0.01) and *** (*p* < 0.001).

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
