# Peer review of "A Pilot Study in Sweden on Efficacy of Benzylpenicillin, Oxytetracycline, and Florfenicol in Treatment of Acute Undifferentiated Respiratory Disease in Calves"

_antibiotics, 2020, doi:10.3390/antibiotics9110736_

Round 1

Reviewer 1 Report

The paper aimed to evaluate the efficacy of benzylpenicillin, oxytetracycline and florfenicol in treatment of acute undifferentiated respiratory disease in calves. This study is of certain value for the rational use of current antibiotics in the treatment of respiratory diseases in calves. However, the experimental design has certain defects, which make the results unconvincing.

  1. As stated in the article that BRD has a multifactorial background that includes infectious agents and environmental factors but also the immunological and general status of the animals. Antibiotics are used only to treat bacterial agents or control secondary infections, and each antibiotic has its own antimicrobial spectrum, however the etiology of the cases included in the study have not been isolated and identified.
  2. The study included a small number of animals and farms and calves were selected for treatment by farm personnel based on visual inspection and rectal temperature which leads to underdiagnosis of calves needing antibiotic therapy. More importantly, the experimental design lacks control group, and the therapeutic effect may not truly reflect the drug's effect.

Author Response

Response to Reviewer 1 Comments

We thank the reviewer for constructive criticism and the relevant issues raised.

Point 1: The reviewer comments that the research design is not applicable.

Response 1: As a general comment the reviewer suggests that the research design must be improved. We fully agree that the design is not optimal but since the study is already carried out it is impossible to change the design now. However, in the manuscript we highlight the limitations of the design on several occasion. Already the title states that this is a pilot study and in the Abstract (lines 27-28 revised manuscript) we acknowledge that the results need to be confirmed in a study with a higher statistical power. Further, in the Discussion there is a long paragraph (lines 199-216 revised manuscript) highlighting the drawbacks of the study design. However, to further clarify the limitations of the study we clarified this a bit more in the Abstract by rephrasing part of the sentence on lines 27-28  to …” but the results need to be confirmed in a more elaborate study with a higher statistical power”.

To evaluate the possibilities of “older” antibiotics is one of the paths suggested to curb emergence of AMR. As stated in Introduction the main objective of the study was to evaluate if bensylpenicillin is an appropriate antibiotic for treatment of calves with undifferentiated BRD. There have been no studies on this for several decades since this “old” antibiotic in most countries is replaced with “newer” substances in treatment of BRD and the interest from pharmaceutical companies to evaluate the possibilities of benzylpenicillin is therefore limited. It is however our belief that on several farms benzylpenicillin is an appropriate treatment for BRD, and with the advantage of a smaller resistance “footprint” than most antibiotics currently used. We therefore decided to evaluate benzylpenicillin in a pilot non-inferiority study to find out whether it had an efficacy comparable to two other antibiotics commonly used for treatment of BRD in calves. With a limited budget we could not afford a more elaborated study with a large number of herds but decided, as a first step, to simply evaluate the efficacy from the farmer’s perspective letting the farmers select which animals that needed treatment and using non-relapse, as perceived by the farmers, and mortality as the main outcome indicators. This is admittedly a simple approach but reflects the routines on most farms. Despite the limitations of the study we therefore think that it is relevant to share the results of this pilot study which indicates that benzylpenicillin performs as well as the other drugs commonly used in routine treatment of BRD on the two farms,. We believe that this information could be an incitement for further and more elaborated work on the topic.

Point 2: The reviewer comments that the methods are not adequately described and that this must be improved.

Response 2: We acknowledge that there might be room for improvements in the description of the methods but since the reviewer gives no concrete examples of poor descriptions it is difficult to comply with this request. Also, none of the two other reviewers considered that there were major deficiencies in the description, but minor issues raised by them have been revised in the M&M section (lines 235-238, 290, revised manuscript).

Point 3: The reviewer comments that the conclusions are not supported by the results.

Response 3: Our main conclusion is that there were no statistically significant difference between the efficacy of benzylpenicillin, oxytetracycline and florfenicol for treatment of BRD in calves and that the response rates of these antibiotics were comparable to what is found for other antibiotics by others. The conclusions are discussed in the context of the limitations of the study and the geographical setting. The only way to improve the support of the conclusions for this manuscript would be to conduct a new study accounting for the limitations of the present and with a higher statistical power - and that is unfortunately not possible.

Point 4: The paper aimed to evaluate the efficacy of benzylpenicillin, oxytetracycline and florfenicol in treatment of acute undifferentiated respiratory disease in calves. This study is of certain value for the rational use of current antibiotics in the treatment of respiratory diseases in calves. However, the experimental design has certain defects, which make the results unconvincing.

1.As stated in the article that BRD has a multifactorial background that includes infectious agents and environmental factors but also the immunological and general status of the animals. Antibiotics are used only to treat bacterial agents or control secondary infections, and each antibiotic has its own antimicrobial spectrum, however the etiology of the cases included in the study have not been isolated and identified.

Response 4: We acknowledge that it is a limitation not to have a subgroup of calves that were sampled for bacteriological culture but that was not possible due to the limited resources available for the study. This is also already acknowledged in the manuscript (lines 214-216 revised manuscript). However, as stated in manuscript, in Sweden Pasteurella multocida is the main species isolated in association with BRD in calves and it is likely that this also was the main species in the two herds studied. This species is susceptible to all three antibiotics studied if not acquired resistance is present – which is uncommon in Sweden as stated in the manuscript (lines 160-162 revised manuscript).

Point 5: The study included a small number of animals and farms and calves were selected for treatment by farm personnel based on visual inspection and rectal temperature which leads to underdiagnosis of calves needing antibiotic therapy. More importantly, the experimental design lacks control group, and the therapeutic effect may not truly reflect the drug's effect.

Response 4: The study was designed as a non-inferiority study comparing the efficacy of three antibiotics since long licensed for treatment of BRD in calves in Sweden and elsewhere. In a non-inferiority study, an untreated control group is not included. Further, in none of the several studies on treatment of BRD in calves cited in the manuscript were control groups included – the basic design of these were similar to the one we used. Also, to use untreated calves in a field study like this would make farmers unwilling to participate because they would be reluctant to leave diseased calves untreated. It would also be unethical and such as study would never be approved by the ethical committees approving clinical research.

Reviewer 2 Report

This is a good study to compare the efficacy of benzylpenicillin, oxytetracycline and florfenicol in treating acute respiratory disease in calves. However, there are some major concerns that need to be addressed.

The primary objective of the study is to compare effectiveness of three different antibiotics in BRD in calves. Why did the authors present the mean response rate (mean response rate of benzylpenicillin, oxytetracycline and florfenicol) for three different time points (at 30 d, 60 d and at slaughter) as a main result of the study (line 22-23; line 95-96; line 121-123)? In other words, three different time points are being compared, NOT three different antibiotics.

In line 225-231 in material and methods, here is the extract: “During 2016, Farm S received 226 2,528 calves emanating from 59 different farms. Of these calves 530 (20.9%) were treated for BRD and 227 43 (1.7%) died or were euthanized before reaching a bodyweight of 100 kg and 40 calves (1.6%) died 228 or were euthanized in the period thereafter and up until slaughter. The same year Farm H received 229 268 calves emanating from one single dairy farm and of these 64 (23.9%) were treated for BRD and 4 230 (1.5%) died or were euthanized before reaching a bodyweight of 100 kg and 8 (3.0%) calves died or 231 were euthanized in the period thereafter”. This will give confusion to the readers since the study enrolled/treated 60 calves each farm. In general, farm S had 530 cases whereas farm H had 64 cases. Why did the study select 60 cases from each farm? Why did not the study select more animals from farm S that had plenty of BRDs.

The authors planned to select 180 BRD cases in calves. Although one farm was dropped out, there were 120 BRD cases. Approximately, there were 40 cases per treatment to determine the differences among treatments. Please reevaluate the statistical power calculation.

The methodology from line 220-224 is not clear enough. Were unweaned calves purchased and brought to a different location and then were they weaned?

This is from line 250. “If farm personnel considered that the clinical response of a calf was unsatisfactory, they could change the treatment to one of the other two regimens”. If the antibiotics would have been switched, how did the authors analyze the efficacy of one antibiotic treatment. The main goal is to compare three different antibiotics.

Line 161: Statement about frequency of BRD treatment is confusing since there is one treatment protocol for each antibiotic.  “PEN: procaine benzylpenicillin 40 mg/kg BW IM, 5 doses 24 H apart (Penovet® vet, Boehringer Ingelheim Animal Health, Copenhagen, Denmark); OTC: oxytetracycline 20 mg/kg BW IM, 2 doses 48 H apart (Engemycin® vet, Intervet AB, Stockholm, Sweden); FLO: florfenicol 20 mg/kg BW, 2 doses 48 H apart (Florselect® vet, Nordvacc Läkemedel AB, Hägersten, Sweden)”

Author Response

Response to Reviewer 2 Comments

We thank the reviewer for constructive criticism and the relevant issues raised.

Point 1: The primary objective of the study is to compare effectiveness of three different antibiotics in BRD in calves. Why did the authors present the mean response rate (mean response rate of benzylpenicillin, oxytetracycline and florfenicol) for three different time points (at 30 d, 60 d and at slaughter) as a main result of the study (line 22-23; line 95-96; line 121-123)? In other words, three different time points are being compared, NOT three different antibiotics.

Response 1: Since there were no statistically significant differences in response rates between the three antibiotics, we considered it “simpler” and less “wordy” to present mean rates in the text – the individual rates are presented in Table 2. However, we agree with the reviewer that it is clearer to present the individual rates also in the text. Therefore, this is now changed in the Abstract (lines 23-25 in the revised manuscript) and in Results (lines 104-109 in the revised manuscript) but we think that using mean response rates in Discussion is appropriate and makes reading “easier” and we have therefor kept the original wording there (lines 133-136 in revised manuscript).

Point 2: In line 225-231 in material and methods, here is the extract: “During 2016, Farm S received 2,528 calves emanating from 59 different farms. Of these calves 530 (20.9%) were treated for BRD and 43 (1.7%) died or were euthanized before reaching a bodyweight of 100 kg and 40 calves (1.6%) died or were euthanized in the period thereafter and up until slaughter. The same year Farm H received 268 calves emanating from one single dairy farm and of these 64 (23.9%) were treated for BRD and 4 (1.5%) died or were euthanized before reaching a bodyweight of 100 kg and 8 (3.0%) calves died or were euthanized in the period thereafter”. This will give confusion to the readers since the study enrolled/treated 60 calves each farm. In general, farm S had 530 cases whereas farm H had 64 cases. Why did the study select 60 cases from each farm? Why did not the study select more animals from farm S that had plenty of BRDs.

Response 2: The data was collected from farm records for year 2016 and thus is some sort of “result” but we decided to put it in the M&M part because it also gives a brief description of the respective farms. We think that information on the number of calves on each farm, the number of BRD-treatments and mortality is important to put the study into context. It can be observed that it is stated in Results (lines 83-85 revised manuscript) that on farm S the collection of 60 consecutive “sick calves” fulfilling the inclusion criteria could be completed already in February-April 2016 whereas it took over a year (Feb 2016- May 2017) to collect 60 calves on the much smaller farm H. When the study started in 2016, we did not have this information and the farmers agreed to participate based on the premise that 60 calves should be enrolled and accepted the ensuing extra workload for that number of calves. It would not have been possible to change the premises once the study was started. In hindsight it might have been possible but the information on the percentage of diseased calves were not available when the study was designed. Also, the study was statistically powered for 60 calves on three separate farms and since “farm” was expected to be a factor of importance for efficacy it would have introduced a statistical bias to include more calves from one farm than from the other.

Point 3: The authors planned to select 180 BRD cases in calves. Although one farm was dropped out, there were 120 BRD cases. Approximately, there were 40 cases per treatment to determine the differences among treatments. Please revaluate the statistical power calculation.

Response 3: Using the sample size calculator Sealed Envelope (https://www.sealedenvelope.com/power/binary-noninferior/) we come to the conclusions presented in the manuscript that in a non-inferiority study 40 calves per treatment at an expected response rate of 85%, a power of 80% and a significance level of 5% makes it possible to detect a difference of more than 20% between treatments. If 60 calves per treatment had been included, it would have been possible to detect a difference of about 15%.

We would be most grateful if the reviewer could indicate what is wrong with these calculations.

Point 4: The methodology from line 220-224 is not clear enough. Were unweaned calves purchased and brought to a different location and then were they weaned?

Response 4: In Sweden it is common practice that unweaned calves (< 8 weeks old) from dairy farms are mediated to farms specialised in raising cattle for slaughter. On arrival to the specialised farm, calves are initially kept on milk substitutes and successively concentrates, and silage are introduced until the calves are weaned at an age of about 8-12 weeks. We have tried to clarify this on lines 235-238 in the revised manuscript.

Point 5: This is from line 250. “If farm personnel considered that the clinical response of a calf was unsatisfactory, they could change the treatment to one of the other two regimens”. If the antibiotics would have been switched, how did the authors analyze the efficacy of one antibiotic treatment. The main goal is to compare three different antibiotics.

Response 5: Once the treatment of a calf was changed from the initial treatment given or if it was retreated following a relapse the calf was considered a “failure” not fulfilling the criteria for a positive RESP30, RESP60 and RESPtot. Thus, we did not evaluate calves for which treatment was changed since they were already “failures”. We have tried to clarify this in M&M, line 290 in the revised manuscript.

Point 6: Line 161: Statement about frequency of BRD treatment is confusing since there is one treatment protocol for each antibiotic.  “PEN: procaine benzylpenicillin 40 mg/kg BW IM, 5 doses 24 H apart (Penovet® vet, Boehringer Ingelheim Animal Health, Copenhagen, Denmark); OTC: oxytetracycline 20 mg/kg BW IM, 2 doses 48 H apart (Engemycin® vet, Intervet AB, Stockholm, Sweden); FLO: florfenicol 20 mg/kg BW, 2 doses 48 H apart (Florselect® vet, Nordvacc Läkemedel AB, Hägersten, Sweden)”.

Response 6: We fail to understand the comment. The regimens used for the three antibiotics are the those recommended for BRD by the respective manufacturer. Thus, to treat a calf for BRD with benzylpenicillin the recommendation from the manufacturer is to give five doses of 40mg/KG IM 24 hours apart to the calf. For OTC, the recommendation is to give two doses of 20mg/KG IM 48 hours apart and for FLO also 20mg/KG IM 48 hours apart. Due to prolonged elimination of the latter two antibiotics it is not necessary to treat the calf daily.

Reviewer 3 Report

Thank you for the opportunity to review this study. The authors evaluate the efficacy of “traditional treatments” to prevent or curb outbreaks of BRD, disease with a high incidence in cattle, in order to avoid the use of antibiotics listed by WHO as critically important for human medicine, that can lead to the creation of resistance.

The results suggest that antibiotics analyzed are effective for treatment of BRD, which is very interesting, but I consider that there are a lot of limitations recognized by the authors themselves, to obtain clear conclusions. For example, the study is developed in a geographic area where some infectious agents of importance in BRD are not present and this reason can explain the good performance of the antibiotics studied. The authors obtain the data of two farms, whose results respect to RESP are very different, perhaps due to farm personnel. I think that the good results of this parameter obtained in the farm H could mask the results.

Author Response

Response to Reviewer 3 Comments

We thank the reviewer for constructive criticism and the relevant issues raised.

Point 1: The reviewer comments that the research design must be improved.

Response 1: As a general comment the reviewer suggests that the research design must be improved. We fully agree that the design is not optimal but since the study is already carried out it is impossible to change the design now. However, in the manuscript we highlight the limitations of the design on several occasion. Already the title states that this is a pilot study and in the Abstract (lines 27-28 revised manuscript) we acknowledge that the results need to be confirmed in a study with a higher statistical power. Further, in the Discussion there is a long paragraph (lines 199-216 revised manuscript) highlighting the drawbacks of the study design. However, to further clarify the limitations of the study we clarified this a bit more in the Abstract by rephrasing part of the sentence on lines 27-28  to …” but the results need to be confirmed in a more elaborate study with a higher statistical power”.

To evaluate the possibilities of “older” antibiotics is one of the paths suggested to curb emergence of AMR. As stated in Introduction the main objective of the study was to evaluate if bensylpenicillin is an appropriate antibiotic for treatment of calves with undifferentiated BRD. There have been no studies on this for several decades since this “old” antibiotic in most countries is replaced with “newer” substances in treatment of BRD and the interest from pharmaceutical companies to evaluate the possibilities of benzylpenicillin is therefore limited. It is however our belief that on several farms benzylpenicillin is an appropriate treatment for BRD, and with the advantage of a smaller resistance “footprint” than most antibiotics currently used. We therefore decided to evaluate benzylpenicillin in a pilot non-inferiority study to find out whether it had an efficacy comparable to two other antibiotics commonly used for treatment of BRD in calves. With a limited budget we could not afford a more elaborated study with a large number of herds but decided, as a first step, to simply evaluate the efficacy from the farmer’s perspective letting the farmers select which animals that needed treatment and using non-relapse, as perceived by the farmers, and mortality as the main outcome indicators. This is admittedly a simple approach but reflects the routines on most farms. Despite the limitations of the study we therefore think that it is relevant to share the results of this pilot study which indicates that benzylpenicillin performs as well as the other drugs commonly used in routine treatment of BRD on the two farms,. We believe that this information could be an incitement for further and more elaborated work on the topic.

Point 2: The reviewer comments that there must be an improvement of the support for the conclusions.

Response 2:

Our main conclusion is that there were no statistically significant difference between the efficacy of benzylpenicillin, oxytetracycline and florfenicol for treatment of BRD in calves and that the response rates of these antibiotics were comparable to what is found for other antibiotics by others. The conclusions are discussed in the context of the limitations of the study and the geographical setting. The only way to improve the support of the conclusions for this manuscript would be to conduct a new study accounting for the limitations of the present and with a higher statistical power - and that is unfortunately not possible.

Point 3: The results suggest that antibiotics analyzed are effective for treatment of BRD, which is very interesting, but I consider that there are a lot of limitations recognized by the authors themselves, to obtain clear conclusions. For example, the study is developed in a geographic area where some infectious agents of importance in BRD are not present and this reason can explain the good performance of the antibiotics studied.

Response 3: Yes, we also recognise that the fact that the study was conducted in Sweden is an important factor. However, we believe that we have been quite clear about that the results are not directly applicable in other settings. On lines 160-166 (revised manuscript) this is discussed in detail but we have also added a sentence to the Abstract to clarify this further (“Due to differences in the panorama of infectious agents and presence of acquired antibiotic resistance, the findings might not be applicable in other geographical areas.” (lines 31-32 revised manuscript)

Point 4: The authors obtain the data of two farms, whose results respect to RESP are very different, perhaps due to farm personnel. I think that the good results of this parameter obtained in the farm H could mask the results.

Response 4: Yes, there were statistically significant differences in both RESP60 and RESPtot between farm S and H. The possible reasons for this are already discussed on lines 174-181 (revised manuscript). The better results on farm H of course improves the mean RESP60 and RESPtot but in our opinion the conclusion of a similar efficacy of the three antibiotics would still be valid. The values of RESP60 and RESPtot for farm S if anything would indicate a better efficacy of benzylpenicillin than of oxytetracycline and florfenicol but there were no statistically significant differences – possibly due to the low statistical power.

Round 2

Reviewer 1 Report

Thank your elaborated responds to my questions, and I have been convinced that although there are some defects in the article, it does not affect the value of his research. However, in order to better ensure the quality of the article, I still have some minor problems as blow:

1.The important foundation and background of this study involved that penicillin resistance in respiratory pathogens from calves is uncommon and multocida is the main species isolated in association with BRD in calves in Sweden; More importantly, multocida also was the main species in the two herds studied. So, relevant epidemiological background should be added at introduction part and provide concerned references.

2. In order for the reader to better understand the research, the flow chart of the research should be supplemented.

3. In line 85-87, “None of these parameters differed significantly (p<0.05) between treatment regimens or between the two farms (Table 1)”. The p<0.05 should be revised to p>0.05,please check it.

Author Response

We thank the reviewer for the relevant comments and have revised the manuscript as outlined below.

Point 1: The important foundation and background of this study involved that penicillin resistance in respiratory pathogens from calves is uncommon and multocida is the main species isolated in association with BRD in calves in Sweden; More importantly, multocida also was the main species in the two herds studied. So, relevant epidemiological background should be added at introduction part and provide concerned references.

Response 1: Unfortunately, we do not have any information on the panorama of infectious agents in the two herds since no bacteriological cultures were performed and can thus not verify that P multocida was the main species in the two herds. However on lines 168-172 in the Discussion the information on the epidemiological background is presented with relevant references (“In Sweden, some infectious agents of importance in BRD elsewhere are not present, for example BVDV and BHV-1 [32 and M. haemolytica, H. somni and M. bovis are less often diagnosed [19, 21]. The bacterial pathogen mainly isolated from calves with BRD in Sweden is P. multocida and antimicrobial resistance to benzylpenicillin, oxytetracycline or florfenicol in this bacterial species is uncommon [19, 20].”). We think that it is necessary to include this information in the Discussion in order to put the results in context. To present the same information already in the Introduction would be to duplicate the information and we would like to avoid that.

We realise that the situation regarding the panorama of infectious agents and antimicrobial resistance in Sweden may be different from that in other countries, which is also highlighted and discussed on lines 164- 176.

However, to further stress that the findings might not be relevant in countries with a different panorama of pathogens we have changed the title to “A pilot study in Sweden on efficacy of benzylpenicillin, oxytetracycline and florfenicol in treatment of acute undifferentiated respiratory disease in calves”.

Point 2: In order for the reader to better understand the research, the flow chart of the research should be supplemented.

Response 2: A flow chart for the outcome of the enrolled calves has been added (Supplement 1) and in addition an overview of the design and elaboration of the study (Supplement 2). We hope that this is what you intended.

Point 3: 3. In line 85-87, “None of these parameters differed significantly (p<0.05) between treatment regimens or between the two farms (Table 1)”. The p<0.05 should be revised to p>0.05, please check it.

Response 3: Thank you for this observation. We apologise for this mistake and have changed < to > were relevant throughout the manuscript (lines: 93, 97, 105, 112, 118,114).

Reviewer 2 Report

The manuscript still needs corrections.

Line 40 to 41: “but in general viral infections of the upper respiratory tract precede bacterial infections of the lower respiratory tract” --- Could you modify this line? It will confuse readers on location of bacteria and viruses in the respiratory tract. In general, viral infections precede bacterial infections in the context of pathogenesis. Upper respiratory tract primarily harbors the bacteria and a few viruses (please refer to microbiota of respiratory tract in cattle). Also, it will flow with the next line in the manuscript.

Line 90: “The relapse rate did not differ between regimens (p<0.05)”. When there is NO significant difference p-value should be > (greater than) 0.05. This symbol needs to be changed throughout the manuscript accordingly.

Line 125 to 128: “The mean response rates (RESP) for all three treatment regimens at 30 days, 60 days and until slaughter for the 117 calves treated were high, 88.9%, 85.5% and 81.2%, respectively, and did not differ between the three regimens (p<0.05)”. In the discussion, the authors did not change as suggested in the previous review. Why has it been confused? There are three drugs. The objective was to compare the efficacies (non-inferiority) of the three different drugs. Taking the average response rates of three antibiotics would not give any meaning. Either these three antibiotics can be compared at each time points, RES-30, RES-60 and RES-slaughter or at RES-slaughter/RES-total.

Author Response

We thank the reviewer for the constructive suggestions and have revised the manuscript as outlined below.

Point 1: Line 40 to 41: “but in general viral infections of the upper respiratory tract precede bacterial infections of the lower respiratory tract” --- Could you modify this line? It will confuse readers on location of bacteria and viruses in the respiratory tract. In general, viral infections precede bacterial infections in the context of pathogenesis. Upper respiratory tract primarily harbors the bacteria and a few viruses (please refer to microbiota of respiratory tract in cattle). Also, it will flow with the next line in the manuscript.

Response 1: We now see the confusion that might arise due to the wording of the sentence and have revised it to : “The pathogenesis and clinical presentation of BRD varies depending on which infectious agents and predisposing factors that are present in a herd, but in general viral infections of the respiratory tract precede secondary bacterial infections”.

Point 2: Line 90: “The relapse rate did not differ between regimens (p<0.05)”. When there is NO significant difference p-value should be > (greater than) 0.05. This symbol needs to be changed throughout the manuscript accordingly.

Response 2: Thank you for this observation. We apologise for this mistake and have changed < to > were relevant throughout the manuscript (lines: 93, 97, 105, 112, 118,114).

Point 3: Line 125 to 128: “The mean response rates (RESP) for all three treatment regimens at 30 days, 60 days and until slaughter for the 117 calves treated were high, 88.9%, 85.5% and 81.2%, respectively, and did not differ between the three regimens (p<0.05)”. In the discussion, the authors did not change as suggested in the previous review. Why has it been confused? There are three drugs. The objective was to compare the efficacies (non-inferiority) of the three different drugs. Taking the average response rates of three antibiotics would not give any meaning. Either these three antibiotics can be compared at each time points, RES-30, RES-60 and RES-slaughter or at RES-slaughter/RES-total.

Response 3: In the Discussion we wanted to avoid a long and complicated sentence to state that the response rates of the three regimens were high and did not differ significantly. Therefore, we used the mean response rates. However, since this apparently may cause some confusion we have changed the sentence to: “The response rates (RESP) at 30 days, 60 days and until slaughter were 90.2%, 87.8% and 80.5%, respectively, for PEN, 90.0%, 85.0% and 85.0%, respectively, for OTC and 86.1%, 82.9% and 77.8%, respectively, for FLO and did not differ between the three regimens (p>0.05).”

Reviewer 3 Report

I appreciate the authors 'answer to my comments and doubts, but it remains a problem for me that  one pathogen only predominates in the disease because the presence of other common pathogens in BRD could modify the response to the proposed antibiotics to treat BRD. The authors indicate that the bacterial pathogen mainly isolated is P multocida. Could the authors confirm the pathogens involved in the disease of these animals? Perhaps,an alternative is to indicate in the title that it is a pilot study in Sweden.

Author Response

We thank the reviewer for constructive criticism and the relevant issues raised.

Point 1: I appreciate the authors 'answer to my comments and doubts, but it remains a problem for me that one pathogen only predominates in the disease because the presence of other common pathogens in BRD could modify the response to the proposed antibiotics to treat BRD. The authors indicate that the bacterial pathogen mainly isolated is P multocida. Could the authors confirm the pathogens involved in the disease of these animals? Perhaps,an alternative is to indicate in the title that it is a pilot study in Sweden.

Response 1: No unfortunately we cannot document the pathogens involved in the two herds studied since no bacteriological cultures were performed. However, we have changed the title according to the suggestion of the reviewer to: “A pilot study in Sweden on efficacy of benzylpenicillin, oxytetracycline and florfenicol in treatment of acute undifferentiated respiratory disease in calves

Round 3

Reviewer 2 Report

The authors have now addressed my concerns. As well, limitations of this pilot study have been clearly stated including in the abstract. There are still some minor errors here and there. Attentive reading multiple times always makes the manuscript close to perfect. 

Line 194: Please correct "about than 20%"

Line 206: Correct "not were"

Reviewer 3 Report

 I appreciate the authors' answer to my comment. I consider that the paper can be published in the present form.